# Dietary Glutamine Supplementation Alleviated Inflammation Responses and Improved Intestinal Mucosa Barrier of LPS-Challenged Broilers

**DOI:** 10.3390/ani12131729

**Published:** 2022-07-04

**Authors:** Bolin Zhang, Qingzhen Zhong, Ning Liu, Peiyong Song, Peng Zhu, Caichao Zhang, Zewei Sun

**Affiliations:** 1Department of Biology and Agriculture, Zunyi Normal College, Ping’an Avenue, Hong Huagang District, Zunyi 563006, China; bolin-zhang@163.com (B.Z.); lxbolin@163.com (N.L.); py66song@126.com (P.S.); zp15761453946@163.com (P.Z.); 17586270132m@sina.cn (C.Z.); 2College of Animal Science and Technology, Jilin Agricultural University, No. 2888, Xincheng Road, Jingyue District, Changchun 130118, China; qingzhenzhong@163.com

**Keywords:** glutamine, lipopolysaccharides, intestine mucosa barrier, intestinal inflammation response, *TLR4*/*FAK*/*MyD88* signaling pathway

## Abstract

**Simple Summary:**

In commercial intense industry, birds have to undergo a series of physical, social and microbial stress. LPS, a structural substance of gram-negative bacterial membrane and an effective immune stimulator for human and animal immune system, can impair growth performance, elevate the production of inflammatory cytokines and destroy the morphology of broilers’ small intestine. Moreover, LPS challenge also can reduce the expression levels of tight junction proteins and ruin the integrity of mucosal barrier of broilers. However, glutamine is considered to be conditionally essential for gut homeostasis and barrier function and maybe a useful strategy to attenuate immunological stress and improve intestine function in response to stressful conditions. Our study showed that 1% Gln supplementation improved the growth performance, alleviated the inflammatory responses and ameliorated the intestinal permeability and the integrity of intestinal mucosa barrier of LPS-challenged broilers.

**Abstract:**

The present study was conducted to investigate the effects of glutamine (Gln) supplementation on intestinal inflammatory reaction and mucosa barrier of broilers administrated with lipopolysaccharide (LPS) stimuli. A total of 120 1-d-old male broilers were randomly divided into four treatments in a 2 × 2 experimental arrangement, containing immune challenge (injected with LPS in a dose of 0 or 500 μg/kg of body weight) and dietary treatments (supplemented with 1.22% alanine or 1% Gln). The results showed that growth performance of broilers intra-abdominally injected with LPS was impaired, and Gln administration alleviated the adverse effects on growth performance induced by LPS challenge. Furthermore, Gln supplementation reduced the increased concentration of circulating tumor necrosis factor-α, interleukin-6 and interleukin-1β induced by LPS challenge. Meanwhile, D-lactic acid and diamine oxidase concentration in plasma were also decreased by Gln supplementation. In addition, the shorter villus height, deeper crypt depth and the lower ratio of villus height to crypt depth of duodenum, jejunum and ileum induced by LPS stimulation were reversed by Gln supplementation. Gln administration beneficially increased LPS-induced reduction in the expression of intestine tight junction proteins such as zonula occludens protein 1 (*ZO-1*), *claudin-1* and *occludin* except for the *ZO-1* in duodenum and *occludin* in ileum. Moreover, Gln supplementation downregulated the mRNA expression of *toll-like receptor 4*, *focal adhesion kinase*, *myeloid differentiation factor 88* and *IL-1R-associated kinase 4* in *TLR4/FAK/MyD88* signaling pathway. Therefore, it can be concluded that Gln administration could attenuate LPS-induced inflammatory responses and improve intestinal barrier damage of LPS-challenged broilers.

## 1. Introduction

LPS, the major structural ingredient of gram-negative bacteria’s cell wall, can initiate intense systemic inflammatory reaction evidenced as the production of inflammatory factors and the altered expressions of genes related to immune system and inflammation [1,2]. Accumulated evidence from previous studies have shown that LPS stimulation can result in comprised growth performance, enhanced production of inflammatory cytokines and dysfunction of intestine structure and function [3,4,5]. Furthermore, in various pathological conditions such as sepsis, inflammation and trauma, it appears to be an increase of intestinal permeability [6,7]. It has been demonstrated that LPS challenge elevated D-lactic acid concentration (D-LA) and diamine oxidase (DAO) activity in serum of broilers administrated with LPS, which are the indicators of intestinal permeability [8]. In addition, intestinal tight junction protein is an indicator of intestinal permeability [9]. Intestinal epithelial cells play a critical role in preserving the wholeness of the epithelial barrier, including the intestinal cytomembrane and tight junctions between enterocytes [10]. These intercellular connections form a barrier for the epithelial cells, which contribute to separating the extracellular fluid at the luminal side from fluid at the serosal side and preventing the interstitial tissues from microbial invasion [11]. Moreover, the special structures making up cell-cell junctions, which contain tight junctions (TJs) and adherens junctions (AJs), affect the effectiveness and stability of the gastrointestinal epithelial barrier [12]. It has been reported that more than 40 different tight junction proteins have been found, which include *occludin*, *claudin-1*, and zonula occludens-1 (*ZO-1*) [13]. However, in the condition of inflammatory or stress responses, the expression of intestinal tight proteins were downregulated. In previous studies, it was demonstrated that LPS challenge reduced the expression levels of tight junction proteins in broilers [14] and in rats [15]. In order to maintain the intestine health and growth performance of broilers at an early age, it is therefore necessary to apply nutritional interventions in broiler production to alleviate the adverse effects induced by LPS administration.

Glutamine (Gln), which serves as a preferential substrate for intestine epithelial cell proliferation and survival under inflammatory conditions, is considered to be conditionally essential for gut homeostasis and barrier function [16,17]. Additionally, it was demonstrated that Gln could be utilized at a high rate by rapid dividing cells such as immune cells and was necessary for lymphocyte proliferation and cytokine production [18]. Although Gln has been considered to be the most abundant amino acid in the circulation, the physiological requirement for Gln may exceed the body’s synthesis capacity under some catabolic stresses [19,20]. Therefore, exogenous Gln addition maybe an effective method to alleviate immunological stress and improve intestine function in response to stressful conditions. However, to our knowledge, the mechanism of Gln regulating the intestinal epithelial barrier in LPS-challenged broilers remains unknown.

Therefore, the objectives of the present study were to investigate the effects of Gln supplementation on growth performance, inflammatory responses and intestinal mucosa barrier of broilers in immunological stress induced by LPS challenge, and furtherly demonstrate the underlying mechanism of Gln supplementation on inflammatory responses and intestinal mucosa barrier of LPS-challenged broilers.

## 2. Materials and Methods

### 2.1. Animal Care and Management

All the procedures including animal care and experiment treatments in the present study were in accordance with guidelines formulated by the Institutional Animal Care and Use Committee of Zunyi Normal College and were approved by the Animal Care and Use Committee.

### 2.2. Diets and Experimental Design

The basal diets were formulated to meet or exceed the recommendations of NRC requirements for broilers. Two diets were formulated with alanine (Ala, 1.22%, control group) and Gln (1.0%, experiment group) on the base of corn- and soybean-based meal, respectively. The dosage of Gln supplementation and Ala as isonitrogenous control were according to previously available studies [18,21]. The raw materials of Ala and Gln were obtained from Shanghai Feiya Technology Development Co., Ltd. (Shanghai, China).

A total of 120 1 d-old male broilers (Arbor Acres Plus) purchased from a local commercial hatchery, were randomly allocated to four treatments in a 2 × 2 factorial design, with five replicates per treatment and six broilers per replicate. The four experimental treatments were as follows: (1) alanine supplementation group, fed the basal diet supplemented with 1.22% Ala and administrated with intraperitoneal injection of sterile saline; (2) Gln supplementation group, fed with diet containing 1% Gln based on the basal diet and receiving intraperitoneal administration of sterile saline; (3) Ala supplementation and LPS group, fed with the basal diet supplemented with 1.22% Ala and administrated with intraperitoneal injection of LPS; (4) Gln supplementation and LPS group, fed with diet containing 1% Gln based on the basal diet and administrated with intraperitoneal injection of LPS. In order to balance the nitrogen level of diet, 1.22% Ala was added into the control diet according to a previous study [22]. The ingredients of diets and nutrition contents were shown in Table 1.

Before LPS challenge, broilers were fed with either Ala or Gln diet for 15 days. Broilers were intraperitoneally injected with LPS solution at a dosage of 500 μg LPS/kg of body weight or the equal volume of 0.9% sterile saline on 8.00 a.m. of d 16 and d 21, respectively [23]. Mean relative humidity was kept approximately at 65%. The LPS (*Escherichia coli* serotype O55: B5, Sigma Chemicals, St. Louis, MO, USA) was dissolved in 0.9% sterile saline solution and the dose of LPS adopted in the present study was according to available findings [3,24]. Broilers were housed in three-level cages (120 × 80 × 60 cm). During the entire experiment, broilers were freely access to feed and water and the diet was suppled in mash form. Broilers were raised in a room equipped with temperature- and light-controlled facilities. The lighting procedure is composed of 23 h light and 1 h darkness. The temperature was maintained between 32 and 34 °C for the initial 3 d and gradually decreased by 2–3 °C weekly until the final temperature was maintained around 26 °C. Moreover, broilers were assessed daily for potential harmful effects of LPS injection on their locomotivity.

### 2.3. Sample Collection

At 21 d of age, 10 broilers (2 broilers per replicate) with a body weight similar with the average body weight of each treatment were randomly selected to collect blood samples within two hours after LPS injection. Blood samples were taken by wing vein puncture and were separately collected into EDTA-coated tubes, then immediately centrifuged at 4 °C 3000× *g* for 10 min to collect supernatants and immediately stored at −20 °C until subsequent analysis. Immediately after blood sampling, the selected broilers were euthanized by cervical dislocation. The abdominal cavity was opened rapidly and the duodenum (from the end of the pylorus to the end of the pancreatic loop), the jejunum (from the end of the pancreatic loop to Meckel’s diverticulum) and ileum (from Meckel’s diverticulum to the cecal junction) were immediately collected and emptied using gentle pressure. The dissected intestinal segments were kept on a pre-cooled stainless steel tray. About 3–5 cm segments of the middle portion of duodenum, jejunum and ileum were collected, opened longitudinally, flushed with ice-cold PBS buffer (pH 7.4) and fixed in chilled neutral-buffered formalin solution for subsequent gut histological measurement. The remaining portions of three intestinal segments mentioned above were used to scrape intestinal mucosa samples using glass microscope slides and immediately frozen in liquid nitrogen until further analysis. All samples were collected within 10 min after killing.

### 2.4. Growth Performance

At 16 and 21 d of age, broilers were weighed after a 12-h feed deprivation period to avoid weight errors induced by the different feed intakes [23]. The total feed consumption and the total weight of birds for each replicate were, respectively, recorded to calculate average feed daily intake (ADFI), average daily gain (ADG) and the ratio of feed to gain (F/G).

### 2.5. Determination of Pro-Inflammatory Cytokines Concentration in Plasma

The concentrations of tumor necrosis factor-α (TNF-α), interleukin-1β (IL-1β) and interleukin-6 (IL-6) in plasma were evaluated using commercially available kits (Nanjing Jiancheng Bioengineering Institute, Nanjing, China) based on the manufacture’s instructions.

### 2.6. Analysis of D-LA and DAO Activity in Plasma

The D-LA concentration in plasma was determined using a D-LA colorimetric assay kit (BioVision Inc., Milpitas, CA, USA) according to the instruction of manufacturer’s recommendations. The activity of DAO in plasma was quantified using commercially available assay kits (Nanjing Jiancheng Bioengineering Institute, Nanjing, China). The operation was according to the manufacture’s protocols.

### 2.7. Intestine Morphology Measurement

Subsequently the intestine segments for morphological were embedded in paraffin after dehydrated through different gradient alcohols. The embedded intestinal tissues were sectioned at 5 μm thickness and then followed by staining with hematoxylin and eosin for measurement. Villus height (from the tip of villus to the junction of villus and crypt) and crypt depth (the depth of invagination between adjacent villi) were measured. Villus height and crypt depth were determined using Image-Pro Plus 6.0. Ten well-oriented, intact villi-crypts units were selected in triplicate for each section.

### 2.8. Total RNA Extraction and Real-Time PCR Measurement

Real-time PCR were used to measure the selected genes ((*toll-like receptor 4, TLR4*), *myeloid differentiation factor 88 (MyD88)*, *ZO-1*, *occludin*, *claudin-1*, *Focal adhesion kinase (FAK)*, *IL-1R associated kinase 4 (IRAK4)*) in duodenum, jejunum and ileum samples, whose total RNA were isolated using RNAiso Plus reagent (catalogue no. 9108, TaKaRa Biotechnology (Dalian) Co., Ltd., Dalian, China) in accordance with the instructions of manufacture. After checking the integrity of RNA by 1% agarose gel, the purity of the obtained RNA was verified by determining the value of OD260/OD280 with by a spectrophotometer (NanoDrop, Thermo Fisher Scientific, Waltham, MA, USA). Total RNA was then reversed into cDNA for further analysis using the PrimeScriptTM RT Master Mix (TaKaRa Biotechnology (Dalian) Co., Ltd., Dalian, China) and real time RT-PCR for the selected genes expression was performed with TB Green Premix Ex Taq (catalogue no. RR420A) according to the instructions of manufacture. PCR programs are programmed as follows: one cycle at 95 °C for 30 s, 40 cycles at 95 °C for 5 s, followed by a step of 60 °C for 30 s. Real-time PCR was carried out using CFX Connect™ real-time PCR detection system CFX Connect (Bio-Rad Laboratories, Hercules, CA, USA.) and each sample were carried out in triplicate. The relative mRNA expression of selected genes to *β-actin* were calculated with the method by Livak and Schmittgen [25]. The relative mRNA expression of each target gene was normalized to those in group of Ala and injection with sterile saline. The primer sequences for the target genes and *β-actin* are shown in Table 2.

### 2.9. Statistical Analysis

The test of normal distribution of all data was carried out using Shapiro-Wilk test and the statistical analyses of all data were conducted by two-way ANOVA to evaluate the main effects (diet (Ala or Gln) and challenge (saline or LPS)) using the GLM procedure of the SAS program (version 8.02, SAS Institute Inc., Cary, NC, USA). Post hoc testing was carried out using LSD multiple comparison if there was a significant (*p* < 0.05) or a trend interaction (0.05 < *p* < 0.1). The pen was defined as an experimental unit for analysis of growth performance, whereas the individual broiler was for the other determining parameters. Results were expressed with means with their standard errors. The *p* < 0.05 was used to indicate statistical significance.

## 3. Results

### 3.1. Growth Performance

As shown in Table 3, compared with saline treatment, LPS challenge decreased ADG and ADFI, whereas LPS increased F/G of broilers (*p* < 0.05). In contrast, Gln supplementation resulted in higher ADG and ADFI (*p* < 0.05) and lower F/G (*p* < 0.05) compared with Ala supplementation group. In addition, F/G of LPS-challenged broilers supplemented with Gln was comparable with those administrated with saline injection and Ala supplementation. No interactions for ADG, ADFI and F/G were observed between LPS challenge and Gln supplementation (*p* > 0.05).

### 3.2. The Concentration of Pro-Inflammatory Cytokines in Plasma

The concentration of TNF-α, IL-1β and IL-6 in plasma of broilers were presented in Figure 1. Compared to saline treatment, the concentration of pro-inflammatory cytokines in plasma of broilers including TNF-α, IL-1β and IL-6, were elevated in response to LPS challenge (*p* < 0.05). However, the increase of TNF-α, IL-1β and IL-6 in plasma induced by LPS exposure were attenuated by Gln administration (*p* < 0.05). A significant interaction for TNF-α and IL-1β were observed between LPS challenge and Gln supplementation (*p* < 0.05). However, there was no interaction for IL-6 between LPS challenge and Gln supplementation (*p* > 0.05).

### 3.3. Intestine Permeability

As shown in Figure 2, LPS challenge significantly increased the D-LA concentration and DAO activity in plasma of LPS-challenged broilers when compared with those treated with saline injection (*p* < 0.05). Compared with Ala supplementation, Gln supplementation resulted in a decrease in D-LA concentration and DAO activity in plasma of broilers (*p* < 0.05). Meanwhile, there are significant interactions for D-LA concentration and DAO activity in plasma between LPS challenge and Gln supplementation (*p* < 0.05).

### 3.4. Intestine Morphology

As shown in Table 4, in comparison with the saline treatment, shorter villus height, deeper crypt depth and lower the ratio of villus height to crypt depth of duodenum, jejunum and ileum of broilers were observed in response to LPS administration (*p* < 0.05). However, Gln reversed the adverse effects of LPS injection on villus height and the ratio of villus height to crypt depth of duodenum, jejunum and ileum (*p* < 0.05). Additionally, the crypt depth of both duodenum and jejunum of LPS challenged-broilers receiving Gln was lower than those supplemented with the control diet (*p* < 0.05). However, the increased crypt depth of ileum of broilers by LPS challenged was not reversed by Gln administration (*p* > 0.05). There is an interaction for the crypt depth of jejunum between LPS challenge and Gln supplementation (*p* < 0.05). However, no interactions for villus height, crypt depth or the ratio of villus height to crypt depth of duodenum, ileum was observed between LPS challenge and Gln supplementation (*p* > 0.05). Meanwhile, there were also no interactions for villus height and the ratio of villus height to crypt depth of jejunum between LPS challenge and Gln supplementation (*p* > 0.05).

### 3.5. mRNA Expressions of Intestine Tight Junction Protein

The results of mRNA level of tight junction protein in duodenum, jejunum and ileum are shown in Table 5. Compared with saline treatment, LPS challenge significantly decreased the mRNA abundances of *claudin-1*, *occludin* and *ZO-1* in mucosa of duodenum, jejunum and ileum of broilers (*p* < 0.05). However, the mRNA expressions of these aforementioned genes, except for *ZO-1* in duodenum and *occludin* in ileum, were reversed with Gln administration (*p* < 0.05). No interactions for mRNA expressions of those genes mentioned above in the mucosa of duodenum, jejunum or ileum were observed between LPS challenge and Gln supplementation (*p* > 0.05).

### 3.6. mRNA Abundances of TLR-4/FAK/MyD88/IRAK4 Signaling Pathway

The results of mRNA levels of TLR-4 signaling factors, including *TLR-4*, FAK, *MyD88* and *IRAK4*, were shown in Table 6. The mRNA abundances of *TLR4*, *FAK*, *MyD88* and *IRAK4* in mucosa of duodenum, jejunum and ileum were enhanced in response to LPS challenge (*p* < 0.05). However, mRNA expressions of these genes mentioned above were reversed by dietary Gln treatment (*p* < 0.05). There were significant interactions for *TLR4* in duodenum and *MyD88* in ileum between LPS challenge and Gln supplementation (*p* < 0.05). However, no interactions for *FAK*, *MyD88* and *IRAK4* in duodenum, *TLR4*, *FAK*, *MyD88* and *IRAK4* in jejunum, and *TLR4*, *FAK* and *IRAK4* in ileum were observed between LPS challenge and Gln supplementation (*p* > 0.05).

## 4. Discussion

In previous studies, LPS challenge resulted in comprised growth performance of broilers [26,27]. In the present study, compared with saline treatment, we found that the growth performance of broilers injected with LPS was depressed, as shown by the reduction of ADG and ADFI and the increased F/G. In consistent with these, a previous study reported that LPS exposure resulted in the decrease of ADG and ADFI [28]. which was mainly attributed to depressed appetite [29], destroyed the intestinal barrier function [3], restrained nutrient absorption [30] and reallocation of nutrients [5,31]. In the condition of immune stress induced by LPS, diets nutrients were directed away from growth, but toward processes in support of various inflammatory immune response and synthesis of various mediators such as acute proteins and cytokines [32]. As a result, the growth performance of broilers by LPS injection was depressed. However, the impaired effects of LPS challenge on ADG and ADFI of broilers were normalized by 1% Gln supplementation. Similar with the results of a previous study, in which it was reported that the growth performance of broilers was improved by 1% Gln supplementation, accompanied by the increased ADFI and ADG and the decreased F/G [33]. Therefore, the results of mentioned above indicated that 1% Gln supplementation could contribute to ameliorating the adverse effects of immune stress induced by LPS challenge on the growth performance of broilers.

Intestinal morphology is one of behavioral markers to evaluate inflammation [34]. In a previous study, in which the negative effects of LPS-challenge on intestine structure was confirmed shown as an increase of crypt depth and a decrease in villus height and the ratio of villus height to crypt depth [28]. Similarly, the same tendency was also found in our study. However, in the present study, diets supplemented with Gln elevated villus height, increased crypt depth, coupled with the increased ratio of villus height to crypt depth in comparison with those administrated by LPS challenge. However, up to now, few studies have been found to study the effects of Gln addition on intestine morphology of broilers challenged by LPS. It has been demonstrated that an increase in villus height and the ration of villus height to crypt depth was observed in broilers under heat stress, which were supplemented with 1% Gln [35]. Additionally, it was reported that Gln supplementation reduced crypt depth, increased villus height and the ration of villus height to crypt depth of broilers with necrotic enteritis challenge [36]. Furthermore, it was demonstrated that improvement of intestinal morphology structure including longer intestinal villus height, deeper crypt depth and higher ratio of villus height to crypt depth contributed to alleviating the stress and ameliorating intestinal barrier functions [37]. The results aforementioned showed that Gln may contribute to protecting intestine morphology and intestinal mucosal of broilers in the state of stress because Gln serves as a sufficient energy substrate for cell proliferation and differentiation [35].

LPS is known to trigger the activity of immune system. Subsequently, the innate immune cells are activated, followed by the release of pro-inflammatory cytokines and anti-inflammatory cytokines [5]. TNF-α, IL-6 and IL-1β, originated from macrophages, are the important cellular messengers that play a major role in diverse inflammatory responses and are highly bound up with the severity of inflammation [38]. In our study, LPS challenge increased the concentration of TNF-α, IL-1β and IL-6 in plasma of broilers in comparison with those receiving saline injection. In consistent with the results of ours, the similar results were also observed by Wu et al. [8] and Chen et al. [39], who reported that the concentration of TNF-α, IL-1β and IL-6 of broilers were elevated in response to LPS challenge. In contrast, in the present study, Gln administration effectively depressed the increase in the contents of TNF-α, IL-1β and IL-6 in plasma induced by LPS challenge. Similarly, a study conducted by Soares et al. [20] showed the similar results. The results of a previous study conducted by Xu et al. [40] also showed that treatment with Gln significantly decreased the levels of TNF-α and IL-6. Moreover, it has been proved that Gln deficiency aggravates the production of proinflammatory cytokines, but Gln supplementation inhibits the inflammatory response in vitro [41]. These therefore suggest that Gln may contribute to alleviating the negative effects of inflammatory responses in response to LPS challenge.

DAO, the class of copper-containing amine oxidases that catalyzes the deamination of histamines through oxidation. The levels of DAO is an indicator of intestinal mucosal barrier function and intestine permeability [42]. DAO is localized in the intestinal epithelial cells and enters the blood circulation through the ruined intestinal barrier. D-LA, a product of fermentation produced by intestinal bacteria, is a sensitive biomarker to reflect intestine injury and to monitor intestinal permeability [9]. In the present study, both DAO and D-LA levels in circulating were elevated by LPS administration, indicating that LPS challenge might seriously impaired intestinal barrier function and damage the intestinal structure, which was in consistent with the results of intestine morphology in the present study mentioned above. Interestingly, the deleterious effect of LPS was markedly reduced by administration of Gln. In agreement with a previous study conducted by Wu et al. [35], who observed that Gln supplementation decreased the D-LA and DAO activity in the plasma of broilers. Indeed, Gln deprivation leads to the impaired paracellular permeability [41]. These results indicated that the intestinal mucosa permeability of broilers supplemented with Gln had somewhat been improved.

The permeability of the intestinal mucosa is influenced by tight junction proteins [42], Claudin family proteins belong to the transmembrane proteins family and are the most important tight junction protein, but ZO family proteins belong to peripheral membrane proteins and are important to tight junction assembly [11]. In the condition of inflammatory responses such as trauma or sepsis, the expression of intestinal tight junction proteins were inhibited [35]. Indeed, in the present study, the mRNA expression of ZO-1, claudin-1 and occludin in duodenum, jejunum and ileum were downregulated. In agreement with the results of ours, the similar results were also found by Gadde et al. [6] and Li et al. [3]. In contrast, Gln supplementation reversed the adverse effects of LPS on intestinal mucosa barriers. Recently, it has been established that Gln are involved in the regulation of gut tight junction proteins [41]. Restoration of tight junctions has been reported after Gln supplementation both in vivo [43] and in vitro [44]. In addition, it was found that Gln deprivation in intestine epithelial cells was associated with a loss of tight junction proteins, whereas Gln addition rescued the phenotype of barrier dysfunction [45]. The results mentioned above indicated that Gln may contribute to preserving the intestinal barrier function of LPS-challenged broilers and furtherly improving the intestinal permeability.

TLR4, which is a signaling receptor of LPS, is thought to be participated in the first immunologic barrier of digestive tract. In addition, TLR4 can initiate the activation of a complex signaling molecules containing various adaptor proteins, kinases and transcriptional factors. It has been proven that TLR4 can trigger the activation of the mitogen-activated protein kinase pathway through MyD88, which is a common adaptor molecule that is recruited towards the Toll/IL-1 receptor domain of the TLRs [40]. FAK, an adaptor protein, plays a critical role in focal adhesion dynamics and initiating the TLR4 signal cascades. Moreover, MyD88 participated in the cytokine release induced by FAK-regulated protein I/II and the increase in the expression of IRAK4 [46]. In our study, LPS stimulation dramatically increased the expression of TLR4, FAK, MyD88 and IRAK4 in duodenum, jejunum and ileum of LPS-challenged broilers compared with those receiving saline injection. In consistent with the results of ours, Guo et al. [46] reported that LPS induces the activation of TLR4 signal transduction cascade, leading to the activation of FAK in intestinal epithelial cells followed by the activation of MyD88 and IRAK4, which contributes to inducing the open of intestinal tight junction. In addition, Luo et al. [47] suggested that TLR4 signaling pathway mediated by LPS challenge maybe involved in the alteration of tight junction proteins via upregulating the transcription and translation of downstream inflammatory factors. These indicated that LPS could result in a quick increase in intestinal permeability and intestinal mucosal destruction by the TLR4/FAK/MyD88 signal transduction axis. To the best of our knowledge, the modulatory effect of Gln on LPS-challenged broilers was not available. Only a previous study about the regulation of Gln intestinal permeability and intestinal mucosa junction tight protein was focused on rats, in which Gln addition increased the expression of TLR4 and MyD88 associated with the morphological destruction of intestinal mucosa and the elevated expression of intestinal tight junction proteins [40]. Furthermore, it has been shown that Gln supplementation may help to reduce the severity of infection, probably associated with the reduction of mucosal cytokine responses and the protection of intestinal barrier function [48]. A study conducted used with piglets infected with Escherichia coli suggested that Gln addition tended to increase the expression of occudin in intestinal mucosa [49]. Additionally, amino acids have important roles in the expression of tight junction proteins. Chen et al. [39] suggested that L-threonine supplementation attenuated inflammatory responses and intestinal barrier damage induced by LPS injection. In the present study, Gln supplementation resulted in a greater effect on suppressing the expression of the related signaling molecules of TLR4/FAK/MyD88/IRAK4 pathway. Furthermore, an increase in the intestinal permeability of LPS-induced mouse was completely prevented in deficient mice of TLR4 or MyD88 or knockdown of FAK [47], indicating that intestinal permeability and intestinal mucosa integrity were regulated by TLR4-dependent activation of FAK/MyD88/IRAK4 signal axis. Therefore, it was reasonable to speculate that Gln supplementation may be contribute to protecting the intestinal mucosa barrier and the prevention of intestinal barrier dysfunction of LPS-challenged broilers through TLR4/FAK/MyD88 signal transduction axis.

## 5. Conclusions

In conclusion, dietary Gln supplementation improved the growth performance of LPS-challenged broilers, alleviated the inflammatory response. Additionally, Gln administration reversed the deleterious effects of LPS on intestinal permeability and the integrity of intestinal mucosa barrier through downregulating the expression of certain molecules involved in TLR4/FAK/MyD88 signaling pathway.

## Figures and Tables

**Figure 1 animals-12-01729-f001:**
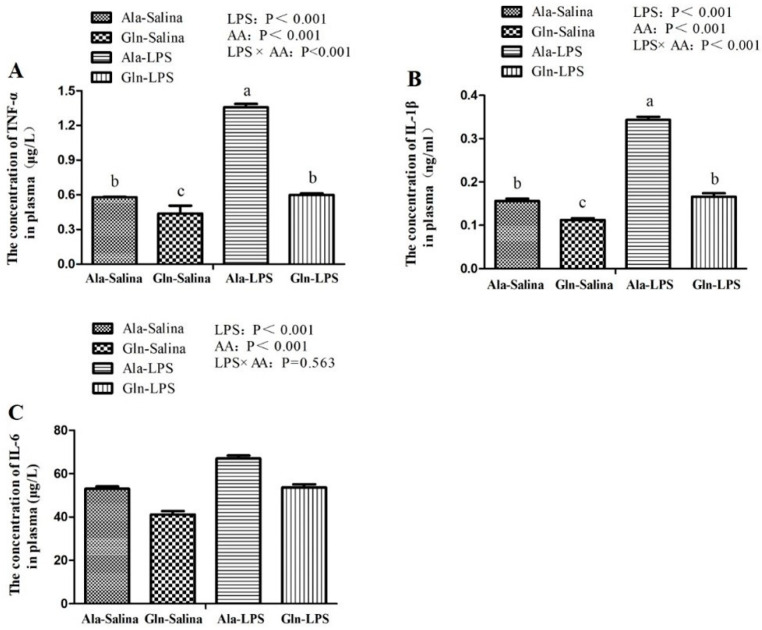
Effects of Gln supplementation on the concentration of tumor necrosis factor-α (TNF-α, (**A**)), interleukin-1β (IL-1β, (**B**)) and interleukin-6 (IL-6, (**C**)) in plasma of broilers following treatment with LPS or saline. Values are means ± SEM *n* = 10 (10 chicken per treatment). The significant difference is indicated by *p* < 0.05.

**Figure 2 animals-12-01729-f002:**
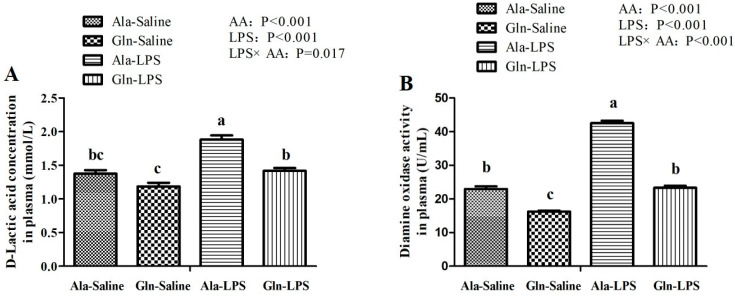
Effects of Gln supplementation on D-lactic acid concentration (**A**) and diamine oxidase activity (**B**) in plasma of broilers following treatment with LPS or saline. Values are means ± SEM *n* = 10 (10 chicken per treatment). The significant difference is indicated by *p* < 0.05.

**Table 1 animals-12-01729-t001:** Ingredients and nutrient content of the basal diets.

Ingredients (g/kg Diet)		Nutrient Content (g/kg Diet)	
Maize	563.0	Crude protein ^‡^	210.8
Wheat bran	51.30	Metabolism energy (MJ/kg)	121.2
Soybean meal	285.0	Calcium (%)	10.00
Corn gluten meal	43.0	Phosphorus (%)	4.50
DL-Methionine	1.50	DL-Methionine (%)	8.60
Phytase	0.40	L-Lysine (%)	10.60
Choline	1.50	Threonine (%)	8.0
Dicalcium phosphate	18.70		
Limestone	12.60		
Salt	1.50		
Soybean oil	16.50		
Vitamin and mineral premix ^†^	5.00		

^†^ Premix per kg diet provided: Vitamin A 12,000 IU; Vitamin D3 2500 IU; Vitamin E 30 mg; menadione 2.8 mg; thiamin 2.21 mg; riboflavin 7.8 mg; nicotinamide 40 mg; Calcium pantothenate 10 mg; pyridoxine·HCl 4 mg; biotin 0.04 mg; folic acid 1.2 mg; Vitamin B12 0.015 mg; Fe 80 mg; Cu 8 mg; Mn 110 mg; Zn 65 mg; I 0.35 mg; Se 0.3 mg. ^‡^ Nutrient content of the diets were the value of measurement.

**Table 2 animals-12-01729-t002:** Sequences used for real-time PCR primers.

Genes	Primers (5′→3′)	Product Size	Gene Bank ^1^
*Occludin*	Sense: GTTACTACTACAGCCCCTTGTTGG	142 bp	NM_205128.1
	Antisense: AGCAGGATGACGATGAGGAA		
*Claudin-1*	Sense: AAGAAGATGCGGATGGCTGT	158 bp	NM_001013611.2
	Antisense: AAGAGGGCTGATCCAAACTCAA		
*ZO-1*	Sense: CTTCAGGTGTTTCTCTTCCTCCTC	131 bp	XM_015278980.2
	Antisense: CTGTGGTTTCATGGCTGGATC		
*TLR4*	Sense: TTCGGTTGGTGGACCTGAATCTTG	114 bp	NM_001030693.1
	Antisense:ACAGCTTCTCAGCAGGCAATTCC		
*FAK*	Sense: CTGTCCTACGCCGACCTCAT	74 bp	NM_205435.1
	Antisense: TTGCTGTCACCCTTATCCTTG		
*MyD88*	Sense: AAGGTGTCGGAGGATGGTGGTC	120 bp	NM_001030962.4
	Antisense: GGAATCAGCCGCTTGAGACGAG		
*IRAK4*	Sense: TGGTTCGCTGCTTGACAGACTTG	98 bp	NM_001030738.1
	Antisense: TGATGCCATTCGCAGTACCTTGAG		
*β-actin*	Sense: ATTGTCCACCGCAAATGCTTC	113 bp	NM_205518.1
	Antisense:AAATAAAGCCATGCCAATCTCGTC		

*ZO-1, Zonula occudens-1; TLR4, Toll-like receptor 4; FAK, focal adhesion kinase; MyD88, myeloid differentiation factor 88; IRAK4, IL-1R-associated kinase 4*. ^1^ Genbank Accession Number.

**Table 3 animals-12-01729-t003:** Effects of dietary Gln supplementation on growth performance of broilers challenged with LPS.

Item	Saline		LPS		SEM	*p* Value		
	Ala	Gln	Ala	Gln		LPS	Gln	LPS × Gln
ADG (g)	46.36	50.14	41.80	45.50	0.586	<0.001	<0.001	0.954
ADFI (g)	72.07	75.48	68.59	70.99	0.774	0.003	0.021	0.661
F/G (g/g)	1.55	1.51	1.65	1.56	0.016	0.009	0.016	0.508

Ala, alanine; Gln, glutamine; ADG, average daily weight gain; ADFI, average daily feed intake; F/G, the ratio of feed to gain. Values are means ± SEM, *n* = 5 (5 replicates per treatment, 6 broilers per replicate). Means in a row with superscripts without a common letter differ, *p* < 0.05.

**Table 4 animals-12-01729-t004:** Effects of dietary Gln supplementation on the small intestine morphology of broilers challenged with LPS.

Item	Saline		LPS		SEM	*p* Value		
	Ala	Gln	Ala	Gln		LPS	Gln	LPS × Gln
Duodenum								
Villus height (μm)	945.2	1016.4	851.8	885.1	15.474	<0.001	0.027	0.401
Crypt depth (μm)	183.9	168.2	196.7	183.3	3.603	0.043	0.039	0.852
H:C (μm/μm)	5.15	6.09	4.38	4.90	0.152	<0.001	0.002	0.339
Jejunum								
Villus height (μm)	880.2	961.9	802.3	835.7	13.178	<0.001	0.004	0.194
Crypt depth (μm)	172.7 ^c^	160.8 ^d^	205.7 ^a^	182.1 ^b^	3.164	<0.001	<0.001	0.018
H:C (μm/μm)	5.09	6.00	3.91	4.59	0.152	<0.001	<0.001	0.417
Ileum								
Villus height (μm)	656.1	700.8	541.1	581.9	11.629	<0.001	<0.001	0.779
Crypt depth (μm)	184.7	181.0	190.7	188.2	1.148	0.002	0.128	0.761
H:C (μm/μm)	3.56	3.87	3.07	3.25	0.059	<0.001	<0.001	0.168

Ala, alanine; Gln, glutamine. H:C, the ratio of villus height to crypt depth. Values are means ± SEM, *n* = 10 (10 chicken per treatment). Means in a row with superscripts without a common letter differ, *p* < 0.05.

**Table 5 animals-12-01729-t005:** Effects of dietary Gln supplementation on mRNA expression of tight junction adhension of intestine of broilers challenged with LPS ^1^.

Item	Saline		LPS		SEM	*p* value		
	Ala	Gln	Ala	Gln		LPS	Gln	LPS × Gln
Duodenum								
*Claudin-1*	1.05	1.50	0.85	1.11	0.069	0.001	0.006	0.323
*Occludin*	1.03	1.18	0.77	0.85	0.042	<0.001	0.019	0.383
*ZO-1*	1.04	1.18	0.71	0.92	0.051	0.006	0.115	0.936
Jejunum								
*Claudin-1*	1.00	1.47	0.41	0.89	0.088	<0.001	<0.001	0.840
*Occludin*	1.01	1.25	0.73	0.91	0.056	0.001	0.013	0.699
*ZO-1*	0.99	1.26	0.65	0.87	0.062	<0.001	0.005	0.748
Ileum								
*Claudin-1*	1.01	1.37	0.63	0.89	0.066	<0.001	<0.001	0.398
*Occludin*	1.07	1.21	0.72	0.84	0.066	0.004	0.234	0.879
*ZO-1*	1.00	1.24	0.87	0.99	0.044	0.011	0.016	0.382

Ala, alanine; Gln, glutamine; *ZO-1, Zonula occludens protein 1.* Values are means ± SEM, *n* = 10 (10 chicken per treatment); ^1^ Broilers were treated with LPS or saline injection.

**Table 6 animals-12-01729-t006:** Effects of dietary Gln supplementation on mRNA expression of TLR4/FAK/MyD88 signaling in intestine of broilers challenged with LPS ^1^.

Item	Saline		LPS		SEM	*p* Value		
	Ala	Gln	Ala	Gln		LPS	Gln	LPS × Gln
Duodenum								
*TLR4*	1.00 ^b^	0.74 ^c^	1.46 ^a^	0.97 ^b^	0.042	<0.001	<0.001	0.003
*FAK*	1.06	0.62	1.21	0.81	0.051	0.037	<0.001	0.785
*MyD88*	1.03	0.77	1.26	1.03	0.058	0.031	0.029	0.852
*IRAK4*	1.08	0.67	1.26	1.05	0.051	0.001	<0.001	0.230
Jejunum								
*TLR4*	1.04	0.56	1.31	0.98	0.066	0.003	0.001	0.479
*FAK*	1.03	0.71	1.21	0.90	0.044	0.013	<0.001	0.988
*MyD88*	1.01	0.53	1.35	0.87	0.064	0.002	<0.001	0.998
*IRAK4*	1.01	0.77	1.26	0.91	0.057	0.038	0.004	0.541
Ileum								
*TLR4*	1.01	0.68	1.63	1.12	0.059	<0.001	<0.001	0.186
*FAK*	1.01	0.88	1.14	0.96	0.025	0.016	0.0011	0.548
*MyD88*	1.05 ^a^	0.48 ^c^	1.11 ^a^	0.84 ^b^	0.042	<0.001	<0.001	0.001
*IRAK4*	1.04	0.57	1.27	0.88	0.072	0.038	0.002	0.749

Ala, alanine; Gln, glutamine; *TLR4, Toll-like receptor 4; FAK, focal adhesion kinase; MyD88, myeloid differentiation factor 88; IRAK4, IL-1R-associated kinase 4*. Values are means ± SEM n = 10 (10 chicken per treatment). Means in a row with superscripts without a common letter differ, p < 0.05; ^1^ Broilers were treated with LPS or saline injection.

## Data Availability

The data presented in this study were available on request from the corresponding author.

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
