# Peer review of "Dietary Glutamine Supplementation Alleviated Inflammation Responses and Improved Intestinal Mucosa Barrier of LPS-Challenged Broilers"

_animals, 2022, doi:10.3390/ani12131729_

Round 1

Reviewer 1 Report

Paper titled “Dietary glutamine supplementation alleviated inflammation responses and improved intestinal mucosa barrier of LPS-challenged broilers” is well designed and interesting study. Only some minor concerns must be elucidated;

Abstract; l. 30-31. GLN reduced the increased concentration of cytokines – please elucidate whether it is still beneficial for birds.

2. l. 38. mRNA abundance is not correct.

3. l. 26-28 vs. that 85-89. There is no mention about Alanine in the last part? As not in the introduction.

4. Regarding the applied LPS challenge; it seems that great attention was put on the gut response to challenge thus, why LPS was not administrated via oral route?

5. Did you measure expression rate of β-actin before considered it as a housekeeping gene for PCR analysis? Why only one housekeeping gene was used for calculation of relative expression of genes?

6. Table 3. Please separate significant differences between groups using different marks (as in case of figure).

7. Why there was a different number of replicates (n=) considered for different analyses?

8. Since H:C is ratio, it is not expressed in arbitrary units.     

Author Response

Abstract; l. 30-31. GLN reduced the increased concentration of cytokines – please elucidate whether it is still beneficial for birds.

Response: Dear reviewer, thanks for your comment. In our study, broilers were challenged by LPS injection through intraperitoneal administration. The results showed that LPS challenge significantly increased the the concentration of TNF-α, IL-1β and IL-6 in plasma of broilers, which   are highly correlated with the severity of inflammation. However, the concentration of TNF-α, IL-1β and IL-6 of broilers were inhibited by Gln addition, which suggested that Gln might contribute to alleviating the negative effects of inflammatory responses in response to LPS challenge. Therefore, we suggested that Gln supplementation is beneficial for birds.

  1. l. 38. mRNA abundance is not correct.

Response: Dear reviewer, thanks for your good advice. “abundance” was replaced with “expression” in Line 42 in the revised manuscript.

  1. l. 26-28 vs. that 85-89. There is no mention about Alanine in the last part? As not in the introduction.

Response: Dear reviewer, alanine was added in diet of the control group. Alanine, a nonessential amino acid, is most appropritate for isonitrogenous control and it is not toxic in this study according to the previous reports (Wang et al, 2008). Besides in the preliminary study, alanine supplemented with 1.22% did not affect food intake, the weight and morphology of small intestine, whole body weight gain or expression of genes in the small intestine (Wang et al, 2008). Additionally, alanine chosen as the an isonitrogenous material was preliminarily accepted in most studies (Xiao et al, 2012; Liu et al, 2008; Yao et al, 2008, 2011).

The role of alanine used in the control diets was to obtain isonitrogenous diets in this study as we mentioned in Lines 115-117 in the first manuscript.

References:

(1) Wang J, Chen L, Li P, Li X, Zhou H, Wang F, et al: Gene expression is altered in piglet small intestine by weaning and dietary glutamine supplementation. J Nutr 2008, 138:1025-1032.

(2) Xiao Y, Wu T, Sun J, Yang L, Hong Q, Chen A, et al: Response to dietary L-glutamine supplementation in weaned piglets: A serum metabolomic comparison and hepatic metabolic regulation analysis. J Anim Sci 2012, 90:4421-4430.

(3) Liu Y, Huang J, Hou Y, Zhu H, Zhao S, Ding B, et al: Dietary arginine supplementation alleviates intestinal mucosal disruption induced by Escherichia coli lipopolysaccharide in weaned pigs. Br J Nutr 2008, 100:552-560

(4) Yao K, Yin Y-L, Chu W, Liu Z, Deng D, Li T, et al: Dietary arginine supplementation increases mTOR signaling activity in skeletal muscle of neonatal pigs. The Journal of nutrition 2008, 138:867-872.

(5) Yao K, Guan S, Li T, Huang R, Wu G, Ruan Z, et al: Dietary L-arginine supplementation enhances intestinal development and expression of vascular endothelial growth factor in weanling piglets. British Journal of Nutrition 2011, 105:703-709.

  1. Regarding the applied LPS challenge; it seems that great attention was put on the gut response to challenge thus, why LPS was not administrated via oral route?

Response: Dear reviewer, the pattern of LPS administration in our study was according to the previous studies (Li, et al., 2015; Zhang et al., 2011; Yang et al., 2021). Indeed, there are some reports about LPS administration via oral route, but the dose of oral LPS administration is 2.0-2.5 mg/kg body weight (Wang, et al., 2021; Reisinger et al., 2020). However, the dose of LPS administration in our study through intraperitoneal injection is 500 μg/kg body weight, which is far lower than oral LPS administration. In addition, the price of LPS purchased from Sigma Chemical Inc. is so expensive. Above the information mentioned above, intraperitoneal injection was chosen in our study.

References:

(1) Li, Y., Zhang, H., Chen, Y.P., Yang, M.X., Zhang, L.L., Lu, Z.X., Zhou, Y.M., Wang, T. Bacillus amyloliquefaciens supplementation alleviates immunological stress and intestinal damage in lipopolysaccharide-challenged broilers. Anim Feed Sci Tech, 2015, 208, 119-131.

(2) Zhang, W., Jiang, Y., Zhu, Q., Gao, F., Dai, S., Chen, J., Zhou, G. Sodium butyrate maintains growth performance by regulating the immune response in broiler chickens. Brit Poultry Sci, 2011, 52, 292-301.

(3) Yang, S., Zhang, J., Jiang, Y., Xu, Y.Q., Jin, X., Yan, S.M., Shi, B.L. Effects of Artemisia argyi flavonoids on growth performance and immune function in broilers challenged with lipopolysaccharide, Anim Biosci, 2021, 34(7): 1169-1180.

(4) Wang, H., Yang, F., Song, Z.W., Shao, H.T., Zhang, M., Ma, Y.B., Yang, F. Influence of Escherichia coli endotoxemia on danofloxacin pharmacokinetics in broilers following single oral administration, Journal of veterinary pharmacology and therapeutics, 2021, 45: 220-225.

(5) Reisinger, N., Emsenhuber, C., Doupover, B., Mayer, E., Schatzmayr, G., Nagl, V. Grenier, B. Endotoxin translocation and gut inflammation are increased in broiler chickens receiving an oral lipopolysaccharide (LPS) bolus during heat stress, 2020, Toxins, 21:622.

  1. Did you measure expression rate of β-actin before considered it as a housekeeping gene for PCR analysis? Why only one housekeeping gene was used for calculation of relative expression of genes?

Response: Dear reviewer, β-action chosen as the housekeeping gene in our study was according to the previous studies (Yang et al., 2021; Li et al., 2015).

References:

(1) Yang, S., Zhang, J., Jiang, Y., Xu, Y.Q., Jin, X., Yan, S.M., Shi, B.L. Effects of Artemisia argyi flavonoids on growth performance and immune function in broilers challenged with lipopolysaccharide, Anim Biosci, 2021, 34(7): 1169-1180.

(2) Li, Y., Zhang, H., Chen, Y.P., Yang, M.X., Zhang, L.L., Lu, Z.X., Zhou, Y.M., Wang, T. Bacillus amyloliquefaciens supplementation alleviates immunological stress and intestinal damage in lipopolysaccharide-challenged broilers. Anim Feed Sci Tech, 2015, 208, 119-131.

  1. Table 3. Please separate significant differences between groups using different marks (as in case of figure).

Response: Dear reviewer, thanks for your suggestion. After reading the latest published article, we find that the letter should be presented with A, B and C. Therefore, we revised it according the instructions of “animals”. The superscripts marked was conducted according to the previous studies (Yang et al., 2021; Wang et al., 2016 a,b ). Besides, superscripts above columns should be listed if P value of “LPS×Gln“ is less than 0.05. So we did not list superscripts when the P value of “LPS×Gln“ is more than 0.05.

References:

(1) Yang, S., Zhang, J., Jiang, Y., Xu, Y.Q., Jin, X., Yan, S.M., Shi, B.L. Effects of Artemisia argyi flavonoids on growth performance and immune function in broilers challenged with lipopolysaccharide, Anim Biosci, 2021, 34(7): 1169-1180.

(2) Wang, W.W., Li, Z., Han, Q.Q., Guo, Y.M., Zhang, B., D`inca, R. Dietary live yeast and mannan-oligosaccharide supplementation attenuate intestinal inflammation and barrier dysfunction induced by Escherichia coli in broilers, 2016a, British journal of nutrition, 1-11

(3) Wang, W.W., Li, Z., Ren, W.L., Yue, Y.S., Guo, Y.M. Effects of live yeast supplementation on lipopolysaccharide-induced inflammatory responses in broilers, 2016b, Poultry science, 0:1-8

  1. Why there was a different number of replicates (n=) considered for different analyses?

Response: Dear reviewer, in Table 1, In our experiment, a total of 120 1d-old male broilers were randomly allocated to four treatments, containing five replicates per treatment and six broilers per replicate. The pen (each replicate) was as an experimental unit for the parameter of growth performance because feed intake recorded as a pen. At the end of our experiment, ten broilers were randomly chosen from each treatment to be treated with LPS injection. Therefore, the individual broilers were the experimental unit for the other parameters. So there was a different number of replicates considered for the growth performance and other determined parameters.

  1. Since H:C is ratio, it is not expressed in arbitrary units.

Response: Dear reviewer, thanks for your suggestion. We have revised this mistake. See details in Table 4 in the revised manuscript. Thanks again!

Reviewer 2 Report

The manuscript was conducted to investigate the effects of dietary Gln on intestine inflammatory responses and intestinal mucosa barrier of LPS-challenged broilers. The results provided some interesting data. Some comments for the manuscript need to be addressed before it is accepted.

1.  There are some irregular spellings in the manuscript as follows, but not limited:

Line 13: “birds are have to……..” in which the word “are” should be deleted.

Line 400: “……directly increased…….”, should be “directly increasing…..”

Line 423: “……be contribute to the protecting the intestinal mucosa barrier………”, should be “……be contribute to protecting the intestinal mucosa barrier………”. 

2. Lines 97-102: This paragraph describes the contents of the diet, so it should be move to next part “2.2 Diets and experimental design”. 

3. Line 132: Sample were obtained “At 21 d of age after LPS injection”. How many hours after LPS injection were the samples taken? 

4. In Figure 1, the letters (a, b, c) on vertical coordinate in Figure 1a, b and c should be deleted. In addition, more information of Figure 1c should be provided, such as superscripts above columns, P values of LPS, Gln and LPSâ…¹Gln. 

5. Table 3 to Table 6: superscripts of means in rows in these tables were missing, except that some in Table 6 were represented. Add the superscript on the means in these tables, please. 

6. TLR4/FAK/MyD88 signaling pathway played significant role in protective effect of Gln on intestinal mucosa barrier of the LPS- challenged broilers. But other signaling pathways, such as Nrf2-ARE, NF-κB/NLRP3, also contributed to the prevention of intestinal barrier dysfunction of LPS-challenged birds. Discussion on these signaling pathways should added in the manuscript, and some studies as following should be cited:

Jin, S.J, Yang, H., Jiao, Y.H., Pang, Q., Wang, Y.J., Wang, M., Shan, A.S., Feng, X.J. Dietary curcumin alleviated acute ileum damage of ducks (Anas platyrhynchos) induced by AFB1 through regulating Nrf2-ARE and NF-κB signaling pathways. Foods. 2021, 10(6), 1370. doi.org/10.3390/foods10061370

Yang, H., Wang, Y.J., Yu, C.T., Jiao, Y.H., Zhang, R.S., Jin, S.J., Feng, X.J. Dietary resveratrol alleviates AFB1-induced ileum damage in ducks via the Nrf2 and NF-κB/NLRP3 signaling pathways and CYP1A1/2 expressions. Agriculture, 2022, 12(1), 54. https://doi.org/10.3390/agriculture12010054.

Author Response

The manuscript was conducted to investigate the effects of dietary Gln on intestine inflammatory responses and intestinal mucosa barrier of LPS-challenged broilers. The results provided some interesting data. Some comments for the manuscript need to be addressed before it is accepted.

  1. There are some irregular spellings in the manuscript as follows, but not limited:

Line 13: “birds are have to……..” in which the word “are” should be deleted.

Response: Dear reviewer, thanks for your good advice. We are sorry for this mistake. We have revised this according to your suggestion. See details in Line 15 in the revised manuscript. Thanks again!

Line 400: “……directly increased…….”, should be “directly increasing…..”

Response: Dear reviewer, thanks for your good advice. We are sorry for this mistake. We have revised this according to your suggestion. See details in Line 394 in the revised manuscript. Thanks again!

Line 423: “……be contribute to the protecting the intestinal mucosa barrier………”, should be “……be contribute to protecting the intestinal mucosa barrier………”.

Response: Dear reviewer, thanks for your suggestion. We have revised this according to your suggestion. See details in Line 424 in the revised manuscript.

  1. Lines 97-102: This paragraph describes the contents of the diet, so it should be move to next part “2.2 Diets and experimental design”.

Response: Dear reviewer, thanks for your suggestion. We have revised it according to your suggestion. See details in Lines 101-106 in the revised manuscript. Thanks again!

  1. Line 132: Sample were obtained “At 21 d of age after LPS injection”. How many hours after LPS injection were the samples taken?

Response: Dear reviewer, blood sample was collected via wing vein puncture within two hours after LPS injection, which was consistence with a previous report by Li et al. (2015). Moreover, the detailed information about blood sampling was added in the revised manuscript. See details in Lines 133-134 in the revised manuscript.

Reference

(1) Li, Y., Zhang, H., Chen, Y.P., Yang, M.X., Zhang, L.L., Lu, Z.X., Zhou, Y.M., Wang, T. Bacillus amyloliquefaciens supplementation alleviates immunological stress and intestinal damage in lipopolysaccharide-challenged broilers. Anim Feed Sci Tech, 2015, 208, 119-131.

  1. In Figure 1, the letters (a, b, c) on vertical coordinate in Figure 1a, b and c should be deleted. In addition, more information of Figure 1c should be provided, such as superscripts above columns, P values of LPS, Gln and LPSâ…¹Gln.

Response: Dear reviewer, thanks for your suggestion. After reading the latest published article, we find that the letter should be presented with A, B and C. Therefore, we revised it according the instructions of “animals”.

The superscripts marked was conducted according to the previous studies (Yang et al., 2021; Wang et al., 2016 a,b ). Besides, superscripts above columns should be listed if P value of “LPS×Gln“ is less than 0.05. So we did not list superscripts when the P value of “LPS×Gln“ is more than 0.05.

References:

(1) Yang, S., Zhang, J., Jiang, Y., Xu, Y.Q., Jin, X., Yan, S.M., Shi, B.L. Effects of Artemisia argyi flavonoids on growth performance and immune function in broilers challenged with lipopolysaccharide, Anim Biosci, 2021, 34(7): 1169-1180.

(2) Wang, W.W., Li, Z., Han, Q.Q., Guo, Y.M., Zhang, B., D`inca, R. Dietary live yeast and mannan-oligosaccharide supplementation attenuate intestinal inflammation and barrier dysfunction induced by Escherichia coli in broilers, 2016a, British journal of nutrition, 1-11

(3) Wang, W.W., Li, Z., Ren, W.L., Yue, Y.S., Guo, Y.M. Effects of live yeast supplementation on lipopolysaccharide-induced inflammatory responses in broilers, 2016b, Poultry science, 0:1-8

  1. Table 3 to Table 6: superscripts of means in rows in these tables were missing, except that some in Table 6 were represented. Add the superscript on the means in these tables, please.

Response: Dear reviewer, thanks for your suggestion. The superscripts marked was conducted according to the previous studies (Yang et al., 2021; Wang et al., 2016 a,b ). Besides, superscripts above columns should be listed if P value of “LPS×Gln“ is less than 0.05. So we did not list superscripts when the P value of “LPS×Gln“ is more than 0.05.

References:

(1) Yang, S., Zhang, J., Jiang, Y., Xu, Y.Q., Jin, X., Yan, S.M., Shi, B.L. Effects of Artemisia argyi flavonoids on growth performance and immune function in broilers challenged with lipopolysaccharide, Anim Biosci, 2021, 34(7): 1169-1180.

(2) Wang, W.W., Li, Z., Han, Q.Q., Guo, Y.M., Zhang, B., D`inca, R. Dietary live yeast and mannan-oligosaccharide supplementation attenuate intestinal inflammation and barrier dysfunction induced by Escherichia coli in broilers, 2016a, British journal of nutrition, 1-11

(3) Wang, W.W., Li, Z., Ren, W.L., Yue, Y.S., Guo, Y.M. Effects of live yeast supplementation on lipopolysaccharide-induced inflammatory responses in broilers, 2016b, Poultry science, 0:1-8

  1. TLR4/FAK/MyD88 signaling pathway played significant role in protective effect of Gln on intestinal mucosa barrier of the LPS- challenged broilers. But other signaling pathways, such as Nrf2-ARE, NF-κB/NLRP3, also contributed to the prevention of intestinal barrier dysfunction of LPS-challenged birds. Discussion on these signaling pathways should added in the manuscript, and some studies as following should be cited:

Response:Dear reviewer, thanks for your suggestion. The objectives of the present study were to investigate the effects of Gln supplementation on growth performance, inflammatory responses and intestinal mucosa barrier of broilers in immunological stress induced by LPS challenge. According to the previous reports, it has been demonstrated that LPS-induced increases in intestinal tight junction protein and intestinal inflammation are regulated by TLR4-dependent activation of the FAK/MyD88signaling pathway (Guo et al. 2015; Luo et al., 2017). Besides, Xu et al (2014) suggested that Gln addition increased the expression of TLR4 and MyD88 associated with the damage of morphology and structure of the intestinal mucosa and the decreased expression of intestine tight junction proteins. Therefore, we could speculate that Gln supplementation might be contribute to protecting the intestinal mucosa barrier and the prevention of intestinal barrier dysfunction of LPS- challenged broilers.

Additionally, it has been demonstrated that Nrf2-ARE pathway is the most important endogenous antioxidant stress pathway. Moreover, up to the best our knowledge, the effects of Nrf2 and NF-κB/NLRP3 signaling pathway on intestine are mostly correlated with ileum. Indeed, it has  been shown that oxidative stress and inflammation are highly related. Therefore, we can study the oxidative stress of broilers under inflammatory reaction through further study. We added some information about the intestinal inflammatory cited from a previous study (Jin et al., 2021). See details in Line 317 in the revised manuscript. Thanks again!

References:

  • Guo, S.H., Nighot, M., Al-sadi, R., Alhmou, T., Nighot, P., Ma, T.Y. Lipopolysaccharide regulation of intestinal yight junction permeability is mediated by TLR4 signal transduction pathway activation of FAK and MyD88, 2015, The journal of immunology, 195 (10): 4999-5010.
  • Luo H, Guo P, Zhou Q. Role of TLR4/NF-κB in damage to intestinal mucosa barrier function and bacterial translocation in rats exposed to hypoxia. PloS one, 2012, 7, e46291.
  • Xu C, Sun R, Qiao X, Xu C, Shang X, Niu W. Protective effect of glutamine on intestinal injury and bacterial community in rats exposed to hypobaric hypoxia environment. World journal of gastroenterology, 2014, 20, 4662-74

Jin, S.J, Yang, H., Jiao, Y.H., Pang, Q., Wang, Y.J., Wang, M., Shan, A.S., Feng, X.J. Dietary curcumin alleviated acute ileum damage of ducks (Anas platyrhynchos) induced by AFB1 through regulating Nrf2-ARE and NF-κB signaling pathways. Foods. 2021, 10(6), 1370. 

Reviewer 3 Report

Dear Authors,

It was a pleasure reviewing your article.

The paper examines Dietary glutamine supplementation could alleviate inflammation responses and improved intestinal mucosa barrier of LPS-challenged broilers. The topic is very relevant since there is a growing demand for poultry production worldwide and theirs' healthy. The introduction points to the essence of the problem team and explains the need for its analysis. The methods have been properly described, results are well presented and data interpretation is appropriate. The findings are thoroughly discussed, and conclusions are justified by the results. I did not find any objective errors, apart from the need to quote the latest literature reports in this area.

Keep up the good work.

Author Response

Dear Authors,

It was a pleasure reviewing your article.

The paper examines Dietary glutamine supplementation could alleviate inflammation responses and improved intestinal mucosa barrier of LPS-challenged broilers. The topic is very relevant since there is a growing demand for poultry production worldwide and theirs' healthy. The introduction points to the essence of the problem team and explains the need for its analysis. The methods have been properly described, results are well presented and data interpretation is appropriate. The findings are thoroughly discussed, and conclusions are justified by the results. I did not find any objective errors, apart from the need to quote the latest literature reports in this area.

Keep up the good work.

Responses: Dear reviewer, thanks for your affirmation of our study. We have made some revisions according to the comments of other reviewers. The revisions were marked in red in the revised manuscript. Thank you again!

Reviewer 4 Report

The manuscript described the effects of a dietary glutamine supplementation on LPS challenged broiler chickens, focussing on biomarkers related to intestinal functions and integrity. The subject is of practical relevance for poultry nutrition and has been addressed before, as correctly cited also by the authors. The manuscript is well written and presented, but the following issues should be considered by the authors:

M&M:  Please explain the rationale why LPS was given by IP injection and why neither the oral route nor a bacterial challenge (for example E.coli) was not considered.

M&M: Please re-visit the text regarding the LPS challenge: it remains difficult to identify how often the animals were challenged (2.2. indicates 2 injections on day 16 and day 21). Moreover, please clearly describe the time-interval after the last LPS challenge the sampling (blood, tissues) as in section 2.4. also a reference to a 12h feed withdrawal period to measure the body weight is mentioned.

Discussion: The authors present a comprehensive reference list, referring to comparable experiments to attenuate an LPS or E. coli challenge in broilers. It is therefore surprising that the only explanation given to describe the beneficial effects of glutamine is its potential (and non-defined) interaction with the TLR4 signalling pathway, the major LPS pathway (see also ref 46). The authors are strongly encouraged to expand the discussion by presenting the similarities and differences in the MoA of glutamine, L-arginine and L-threonine (and if possible, even other feed additives) on LPS-induces intestinal functions, inflammation and barrier integrity in broilers.   

Author Response

The manuscript described the effects of a dietary glutamine supplementation on LPS challenged broiler chickens, focusing on biomarkers related to intestinal functions and integrity. The subject is of practical relevance for poultry nutrition and has been addressed before, as correctly cited also by the authors. The manuscript is well written and presented, but the following issues should be considered by the authors:

M&M:  Please explain the rationale why LPS was given by IP injection and why neither the oral route nor a bacterial challenge (for example E.coli) was not considered.

Response: Dear reviewer, the pattern of LPS administration in our study was according to the previous studies (Li, et al., 2015; Zhang et al., 2011; Yang et al., 2021). Indeed, there are some reports about LPS administration via oral route, but the dose of oral LPS administration is 2.0-2.5 mg/kg body weight (Wang, et al., 2021; Reisinger et al., 2020). However, the dose of LPS administration in our study through intraperitoneal injection is 500 μg/kg body weight, which is far lower than oral LPS administration. In addition, the price of LPS purchased from Sigma Chemical Inc. is so expensive. Above the information mentioned above, intraperitoneal injection was chosen in our study.

References:

(1) Li, Y., Zhang, H., Chen, Y.P., Yang, M.X., Zhang, L.L., Lu, Z.X., Zhou, Y.M., Wang, T. Bacillus amyloliquefaciens supplementation alleviates immunological stress and intestinal damage in lipopolysaccharide-challenged broilers. Anim Feed Sci Tech, 2015, 208, 119-131.

(2) Zhang, W., Jiang, Y., Zhu, Q., Gao, F., Dai, S., Chen, J., Zhou, G. Sodium butyrate maintains growth performance by regulating the immune response in broiler chickens. Brit Poultry Sci, 2011, 52, 292-301.

(3) Yang, S., Zhang, J., Jiang, Y., Xu, Y.Q., Jin, X., Yan, S.M., Shi, B.L. Effects of Artemisia argyi flavonoids on growth performance and immune function in broilers challenged with lipopolysaccharide, Anim Biosci, 2021, 34(7): 1169-1180.

(4) Wang, H., Yang, F., Song, Z.W., Shao, H.T., Zhang, M., Ma, Y.B., Yang, F. Influence of Escherichia coli endotoxemia on danofloxacin pharmacokinetics in broilers following single oral administration, Journal of veterinary pharmacology and therapeutics, 2021, 45: 220-225.

(5) Reisinger, N., Emsenhuber, C., Doupover, B., Mayer, E., Schatzmayr, G., Nagl, V. Grenier, B. Endotoxin translocation and gut inflammation are increased in broiler chickens receiving an oral lipopolysaccharide (LPS) bolus during heat stress, 2020, Toxins, 21:622.

M&M: Please re-visit the text regarding the LPS challenge: it remains difficult to identify how often the animals were challenged (2.2. indicates 2 injections on day 16 and day 21). Moreover, please clearly describe the time-interval after the last LPS challenge the sampling (blood, tissues) as in section 2.4. also a reference to a 12h feed withdrawal period to measure the body weight is mentioned.

Response: Dear reviewer, as illustrated in our study in Lines 119-121 in the revised manuscript, broilers were injected twice. On day 16 and day 21 of the experiment, broilers were separately intraperitoneally injected with LPS solution at a dosage of 500 μg LPS/kg body weight on 8:00 am after 12 h feed withdraw. The time interval of the injection and 12 h feed withdraw were conducted in accordance with our previous study (Zhang et al., 2020). The reference was added in Line 120 and 150 in the revised manuscript.

Blood sample was collected via wing vein puncture within two hours after LPS injection, which was consistence with a previous report by Li et al. (2015). Moreover, the detailed information about blood sampling was added in the revised manuscript. See details in Lines 133-134 in the revised manuscript. Thanks again!

Reference

(1) Li, Y., Zhang, H., Chen, Y.P., Yang, M.X., Zhang, L.L., Lu, Z.X., Zhou, Y.M., Wang, T. Bacillus amyloliquefaciens supplementation alleviates immunological stress and intestinal damage in lipopolysaccharide-challenged broilers. Anim Feed Sci Tech, 2015, 208, 119-131.

(2) Zhang, B.L., Yang, Q., Song, P.Y., Xie, Y.X., Wang, Q.R., Sun, Z.W. Effects of dietary L-glutamine supplementation on plasma biochemical parameters, immune performance, intestinal inflammatory factors expression and mucosal immune of broilers challenged by lipopolysaccharide, Chinese journal of animal nutrition, 32(6):2611-2623. (in Chinese)

Discussion: The authors present a comprehensive reference list, referring to comparable experiments to attenuate an LPS or E. coli challenge in broilers. It is therefore surprising that the only explanation given to describe the beneficial effects of glutamine is its potential (and non-defined) interaction with the TLR4 signalling pathway, the major LPS pathway (see also ref 46). The authors are strongly encouraged to expand the discussion by presenting the similarities and differences in the MoA of glutamine, L-arginine and L-threonine (and if possible, even other feed additives) on LPS-induces intestinal functions, inflammation and barrier integrity in broilers.  

Response: Dear reviewer, thanks for your suggestion. We searched the recent published papers, however, articles about glutamine supplementation on LPS-induced intestinal barrier integrity of broilers were not available. It was reported that glutamine could improve intestinal integrity and thus barrier function. Besides, glutamine supplementation may be beneficial for individuals with an impaired gut permeability by enhancing the expression of tight junction proteins (Coeffier, M. et al, 2009).

It has been shown that Gln supplementation is beneficial in mitigating the severity of infection, probably by reducing the mucosal cytokine responses and by preserving the intestinal barrier function (Sukhotnik et al.2007). A study conducted used with piglets infected with Escherichia coli suggested that Gln addition tended to increase the expression of occudin in intestinal mucosa (Ewaschuk et al., 2011). Additionally, amino acids have important roles in the expression of tight junction proteins. Chen et al (2018) suggested that L-threonine supplementation attenuated inflammatory responses and intestinal barrier damage induced by LPS injection. All these information were added in Lines 410-417 in the revised manuscript.

References:

(1) Coeffier, M., Marion-letellier, R., Dechelotte, P. Potential for amino acids supplementation during inflammatory bowel diseases. Inflammatory bowel disease, 2010, 16(3): 518-524.

(2) Sukhotnik I, Khateeb K, Mogilner J, Helou H, Lurie M, Coran A, Shiloni E. Dietary glutamine supplementation prevents mucosal injury and modulates intestinal epithelial restitution following ischemia-reperfusion injury in the rat. Digestive diseases sciences, 2007, 52, 1497-504.

(3) Ewaschuk JB, Murdoch GK, Johnson IR, Madsen KL, Field CJ. Glutamine supplementation improves intestinal barrier function in a weaned piglet model of Escherichia coli infection. British Journal of Nutrition, 2011, 106, 870-7.

(4) Chen Y, Zhang H, Cheng Y, Li Y, Wen C, Zhou Y. Dietary l-threonine supplementation attenuates lipopolysaccharide-induced inflammatory responses and intestinal barrier damage of broiler chickens at an early age. The British journal of nutrition, 2018, 119, 1254-62.
